- Polarity and direction dependence of energetic cross-frontal eddy transport in the Southern
- Ocean's Pacific sector
- Huimin WANG<sup>1</sup>, Lingqiao CHENG<sup>1,5\*</sup>, Erik BEHRENS<sup>2</sup>, Zhuang CHEN<sup>1,4,5</sup>, Jennifer DEVINE<sup>3</sup>, Guoping
- ZHU<sup>4,5,6,7</sup>
- <sup>1</sup>College of Oceanography and Ecological Science, Shanghai Ocean University, Shanghai, China
- <sup>2</sup>Earth Sciences New Zealand, Wellington, New Zealand
- <sup>3</sup>Earth Sciences New Zealand, Nelson, New Zealand
- 4College of Marine Living Resource Sciences and Management, Shanghai Ocean University, Shanghai, China
- <sup>5</sup>Center for Polar Research, Shanghai Ocean University, Shanghai, China
- <sup>6</sup>Polar Marine Ecosystem Group, The Key Laboratory of Sustainable Exploitation of Oceanic Fisheries Resources,
- Ministry of Education, Shanghai, China
- <sup>7</sup>International Research Center for Marine Biosciences at Shanghai Ocean University, Ministry of Science and
- Technology, China
- \* Correspondence: Lingqiao Cheng (lqcheng@shou.edu.cn)

#### Abstract:

15

16 Cross-frontal mesoscale eddies mediate meridional heat and mass transport across the Antarctic Circumpolar 17 Current fronts. Yet their spatiotemporal characteristics and dynamical impacts in the Southern Ocean's Pacific 18 sector remain inadequately quantified. Utilizing 23 years (2000-2022) of satellite altimetry and Argo float data, we 19 reveal, for the first time, a pronounced polarity and direction dependence in cross-frontal eddy (CFE) abundance, 20 energetics, and interactions with jets. Equatorward-propagating cyclonic eddies (CEs) dominate CFE activity (36% of total), exhibiting higher eddy kinetic energy (EKE, in terms of total EKE in eddy interiors,  $EKE_T$ ), longer 21 propagation distances, and stronger nonlinearity than other types, followed by poleward-moving anticyclonic 22 23 eddies (AEs, 28%). These two dominant directional groups primarily drive the significant increase in the overall CFEs'  $EKE_T$ : CEs at  $(1.98 \pm 1.53) \times 10^6$  m<sup>4</sup> s<sup>-2</sup> yr<sup>-1</sup> (excluding the anomalously low 2017 value) and AEs at  $(1.58 \pm 1.53) \times 10^6$  m<sup>4</sup> s<sup>-2</sup> yr<sup>-1</sup> (excluding the anomalously low 2017 value) and AEs at  $(1.58 \pm 1.53) \times 10^6$  m<sup>4</sup> s<sup>-2</sup> yr<sup>-1</sup> (excluding the anomalously low 2017 value) and AEs at  $(1.58 \pm 1.53) \times 10^6$  m<sup>4</sup> s<sup>-2</sup> yr<sup>-1</sup> (excluding the anomalously low 2017 value) and AEs at  $(1.58 \pm 1.53) \times 10^6$  m<sup>4</sup> s<sup>-2</sup> yr<sup>-1</sup> (excluding the anomalously low 2017 value) and AEs at  $(1.58 \pm 1.53) \times 10^6$  m<sup>4</sup> s<sup>-2</sup> yr<sup>-1</sup> (excluding the anomalously low 2017 value) and AEs at  $(1.58 \pm 1.53) \times 10^6$  m<sup>4</sup> s<sup>-2</sup> yr<sup>-1</sup> (excluding the anomalously low 2017 value) and AEs at  $(1.58 \pm 1.53) \times 10^6$  m<sup>4</sup> s<sup>-2</sup> yr<sup>-1</sup> (excluding the anomalously low 2017 value) and AEs at  $(1.58 \pm 1.53) \times 10^6$  m<sup>4</sup> s<sup>-2</sup> yr<sup>-1</sup> (excluding the anomalously low 2017 value) and AEs at  $(1.58 \pm 1.53) \times 10^6$  m<sup>4</sup> s<sup>-2</sup> yr<sup>-1</sup> (excluding the anomalously low 2017 value) and AEs at  $(1.58 \pm 1.53) \times 10^6$  m<sup>4</sup> s<sup>-2</sup> yr<sup>-1</sup> (excluding the anomalously low 2017 value) and AEs at  $(1.58 \pm 1.53) \times 10^6$  m<sup>4</sup> s<sup>-2</sup> yr<sup>-1</sup> (excluding the anomalously low 2017 value) and  $(1.58 \pm 1.53) \times 10^6$  m<sup>4</sup> yr<sup>-1</sup> (excluding the anomalously low 2017 value) and  $(1.58 \pm 1.53) \times 10^6$  m<sup>4</sup> yr<sup>-1</sup> (excluding the anomalously low 2017 value) and  $(1.58 \pm 1.53) \times 10^6$  m<sup>4</sup> yr<sup>-1</sup> (excluding the anomalously low 2017 value) and  $(1.58 \pm 1.53) \times 10^6$  m<sup>4</sup> yr<sup>-1</sup> (excluding the anomalously low 2017 value) and  $(1.58 \pm 1.53) \times 10^6$  m<sup>4</sup> yr<sup>-1</sup> (excluding the anomalously low 2017 value) and  $(1.58 \pm 1.53) \times 10^6$  m<sup>4</sup> yr<sup>-1</sup> (excluding the anomalously low 2017 value) and  $(1.58 \pm 1.53) \times 10^6$  m<sup>4</sup> yr<sup>-1</sup> (excluding the anomalously low 2017 value) and  $(1.58 \pm 1.53) \times 10^6$  m<sup>4</sup> yr<sup>-1</sup> (excluding the anomalously low 2017 value) and  $(1.58 \pm 1.53) \times 10^6$  m<sup>4</sup> yr<sup>-1</sup> (excluding the anomalously low 2017 value) and  $(1.58 \pm 1.53) \times 10^6$  m<sup>4</sup> yr<sup>-1</sup> (excluding the anomalously low 2017 value) and  $(1.58 \pm 1.53) \times 10^6$  m<sup>4</sup> yr<sup>-1</sup> (excluding the anomalously low 2017 24 0.74) ×  $10^6$  m<sup>4</sup> s<sup>-2</sup> yr<sup>-1</sup>. Specifically, complete CFEs (experience pre-crossing, crossing, and post-crossing phases) 25 are responsible for these trends, distinct from non-CFEs, partial or transient CFEs, which show no trend. During 26 27 frontal crossing, EKET enhances in equatorward CEs and poleward AEs but diminishes in poleward CEs and equatorward AEs, explaining the two former types' capacity for long-distance propagation and energetic behaviors. 28 29 The intensified CEs carry cold, fresh southern waters equatorward, while AEs transport warm, salty northern waters 30 poleward. These cross-frontal exchanges mitigate thermohaline gradients between interfrontal zones while 31 enhancing local mesoscale available potential energy. We conclude that CFEs serve dual climatic roles, in 32 mediating meridional energy transport while dynamically stabilizing the ACC against strengthening winds.

- Keywords: Cross-frontal eddies; Eddy kinetic energy, Spatiotemporal characteristics; Energy transfer; Antarctic
- Circumpolar Current; the Pacific sector of the Southern Ocean

3

#### 1 Introduction

Mesoscale eddies are ubiquitous in the Southern Ocean (SO), play a vital role in the zonal and meridional transport of quantities including heat and momentum across the Antarctic Circumpolar Current (ACC), and also 37 38 influence the uptake of heat and carbon dioxide from the atmosphere (Moreau et al., 2017; Patel et al., 2019; Sallée 39 et al., 2008; Sokolov and Rintoul, 2007) and the transport and connectivity of marine species (e.g., Duan et al., 2021; Zhu et al., 2025). The ACC comprises multiple zonal fronts, where oceanic jets exit. At these fronts, 40 41 mesoscale activity is enhanced, with higher eddy kinetic energy (EKE) and more frequent eddy generation and 42 dissipation (Barthel et al., 2017; Gille, 2012; Hughes, 2012; Hughes and Ash, 2001; Morrow et al., 1994; Sokolov 43 and Rintoul, 2002). In turn, eddies impact the fronts' structure, intensity, and location. For instance, eddies may 44 accelerate the jets, and cyclonic (anticyclonic) eddies cause the equatorward (poleward) deviation of frontal 45 meanders in some cases (Chapman et al., 2020; Duan et al., 2016; Frenger et al., 2015; Sprintall, 2003). These 46 interactions between mesoscale eddies and oceanic fronts can shape local thermohaline structures, exert profound 47 influences on large-scale circulation and vertical flux processes. They also have tremendous implications for the 48 redistribution and survival of marine species and the stability of the climate system. 49 In the SO, the transition from the warm subtropical waters to the cold Antarctic waters is not smooth but concentrated along a series of fronts (Deacon, 1937), often corresponding to the locations of narrow, high-speed 50 51 currents known as "jets" (Sokolov and Rintoul, 2002, 2007). These fronts delineate the boundaries of distinct water 52 masses, each with unique environmental characteristics (Orsi et al., 1995). The existence of fronts hinders 53 meridional exchanges of heat and tracers (Chapman and Sallée, 2017; Naveira Garabato et al., 2011; Thompson 54 and Sallée, 2012a). At the same time, eddies enable cross-frontal transport and serve as primary carriers for 55 meridional water mass properties, including heat (De Szoeke and Levine, 1981; Foppert et al., 2017). Practically, 56 Cross-frontal eddies (CFEs) must overcome intense geostrophic shear to achieve material transport and render their 57 dynamical contributions to meridional transport, which is significantly more pronounced compared to other eddy 58 types (Thompson and Sallée, 2012b). 59 Eddies' capability of trapping materials and achieving long-distance cross-frontal transport helps in mitigating 60 sharp meridional hydrographic gradients, facilitating new water formation and carbon transport, and also enhancing 61 subsurface temperature extremes in the SO. Holte et al. (2013) presented that cross-frontal exchanges by eddies can 62 penetrate strong potential vorticity gradients associated with the Subantarctic Front (SAF) and facilitate the 63 downstream evolution of Subantarctic Mode Water by transporting cold, low-salinity water across the ACC from the Polar Front Zone (Holte et al., 2013). In a study of a cold eddy in the southwest Indian Ocean, Swart et al. 64

65 (2008) found that the eddy displaced temperature and salinity anomalies by 1.5° towards lower latitudes. This single eddy contributed 2.5% of the annual northward flux of Antarctic Surface Water in the southwest Indian 66 sector (Swart et al., 2008). In addition, eddies induce carbon transport across the ACC, which alters the carbon 67 68 properties and budget of the Subantarctic Zone waters (Moreau et al., 2017). Patel et al. (2019) proposed that about 69 21% of the heat transported across the SAF to the Subantarctic Zone south of Tasmania is achieved by cyclonic eddies. He et al. (2023) demonstrated that nearly half of the observed subsurface temperature extremes in the SO 70 71 occur within eddies; CFEs act as a source of extremely high-temperature events on the cold side of the ACC and 72 extremely low-temperature events on the warm side; These temperature extremes eventually impact marine 73 organisms and ecosystems. For instance, Electrona calserbgi in the high-latitude Antarctic region may be 74 transported across the fronts from the Argentine Basin by the poleward eddy activity (Saunders et al., 2017; Zhu et 75 al., 2025). 76 Eddies in the SO can moderate the ACC's response to surface wind forcing changes, namely the "eddy saturation" hypothesis (Hallberg and Gnanadesikan, 2001, 2006; Straub, 1993). Reanalysis of data since 1972 show 77 78 an increasing trend in wind stress (associated with a positive trend of Southern Annular Mode) over the Pacific 79 sector that dominates the basin-wide wind stress variability, driving enhanced eddy activity responses in this sector, with EKE intensifying at a rate of  $14.9 \pm 4.1$  m<sup>4</sup> s<sup>-2</sup> per decade (Duan et al., 2016; Hogg et al., 2015; Menna et al., 80 81 2020; Morrow et al., 2010). Recent work by Zhang et al. (2021) demonstrates that EKE intensification is not 82 spatially homogeneous in the SO but concentrated south of New Zealand and downstream of the Campbell Plateau 83 in the Pacific sector. This localized enhancement likely stems from the release of available potential energy stored 84 in tilted isopycnals, thereby mitigating the eastward flow in the ACC, which has significantly intensified between 85 48°S and 58°S mainly due to buoyancy forcing (Shi et al., 2021). Mesoscale energy gain from mean flows is 86 achieved through baroclinic (primary) and barotropic (secondary) pathways (Fu et al., 2023). Regarding 87 topographic effects, previous studies have established that interactions between ACC and seafloor topography 88 intensify oceanic eddy mixing by enhancing downstream baroclinic shear. This process enhances eddy generation and increases EKE downstream of major topographic features (Frenger et al., 2015; Morrow et al., 1992; Park et al., 89 90 1993; Thompson and Sallée, 2012a). Consequently, ACC frontal jets with strong geostrophic characteristics 91 experience mesoscale eddy modulation near prominent topographies (Kim and Orsi, 2014; Thompson et al., 2010). 92 Despite extensive research on basin-scale EKE modulations and case studies of CFE transport, and the 93 well-established asymmetric eddy distribution on both sides of the ocean fronts, fundamental questions remain regarding how eddy-jet interactions vary based on eddy characteristics and directional approach in the SO. 94

https://doi.org/10.5194/egusphere-2025-5038 Preprint. Discussion started: 21 October 2025 © Author(s) 2025. CC BY 4.0 License.

Specifically, it is essential to understand: (1) the polarity and direction preferences of CFEs during frontal crossing; (2) the magnitude and pattern of kinetic energy change within eddies following frontal crossing; (3) the resultant hydrographic property redistribution achieved by CFEs in the interfrontal zones. Motivated by these research gaps, we conducted a systematic assessment of cross-frontal mesoscale eddies in the Pacific sector to elucidate their role in regional ocean dynamics and hydrographic redistribution. Utilizing 23 years (2000–2022) of satellite altimetry data, we characterize the spatiotemporal variability, *EKE* patterns, and eddy-jet interactions of CFEs in the Pacific sector. We complement these analyses with Argo (Array for Real-time Geostrophic Oceanography) float profiles to quantify normalized hydrographic differences between cyclonic and anticyclonic eddies within the interfrontal zones. These approaches aim to improve our understanding of the dynamic characteristics of CFEs and their role in mediating meridional transport across the ACC in this sector.

#### 2 Data and methods

#### 2.1 Data

This study focuses on the SO's Pacific sector between 150°E-110°W and 35°S-80°S. Prominent topographic features within this region include the Campbell Plateau, Pacific-Antarctic Ridge, and Udintsev Fracture Zone (Figure 1). We utilized the gridded satellite altimeter data for eddy detection and tracking. This dataset is merged from multiple satellites and provided by the Copernicus Marine Service (Global Ocean Gridded L4 Sea Surface Heights And Derived Variables at https://marine.copernicus.eu/). It includes daily Sea Level Anomaly (SLA) and sea surface geostrophic velocity anomalies (u', v') data during 2000 and 2022 with a spatial resolution of 0.25°×0.25°. The SLA is the sea surface height anomaly relative to the mean sea surface from 1993 to 2012. u', v' were calculated from SLA based on the geostrophic relation:

115 
$$u' = -\frac{g}{f} \frac{\partial SLA}{\partial y}, v' = \frac{g}{f} \frac{\partial SLA}{\partial x}, \qquad (1)$$

where g is the acceleration of gravity (m s<sup>-2</sup>), f is the Coriolis parameter (s<sup>-1</sup>), x and y are the zonal and meridional distances (m), respectively.

The frontal data used in this study were sourced from Park et al. (2019). In their frontal identification process, the climatological positions of the ACC fronts were determined using the CNES-CLS18 mean dynamic topography (Mulet et al., 2012) derived from satellite altimetry. Then, subsurface temperature data (2001–2017) from Argo floats were employed to validate the satellite-derived frontal positions. In addition, high-quality hydrographic data from two CTD (Conductivity-Temperature-Depth) surveys conducted in 2016 and 2017 further verified the ACC frontal structures. By integrating these datasets, Park et al. (2019) produced the most updated mapping of the ACC fronts. As shown in Figure 1, from north to south, the key fronts include the northern boundary of ACC (NB), the Subantarctic Front (SAF), the Polar Front (PF), the Southern Antarctic Circumpolar Current Front (SACCF), and the southern boundary of the ACC (SB).

Furthermore, we utilized a total of 1,165 quality-controlled Argo profiles (0–2000 m; http://www.argo.net) observed in eddies to normalize the internal temperature and salinity properties in cyclonic eddies (CEs) and anticyclonic eddies (AEs) in the interfrontal zones. The normalized potential temperature ( $\theta$ ) and salinity (S) in the radius direction were derived by matching Argo profiles to eddies based on the ratio of their distance from the eddy center to the eddy radius (R). The average intervals were 0.006R for CEs and 0.004R for AEs, respectively, based on sample density. Due to limited Argo float coverage, temporal variability (e.g., interannual and seasonal) was not considered in this normalizing process, and the SB-SACCF region and those south of SB were omitted. This

7

- analysis focuses only on the northern inter-frontal zones of SAF-NB, PF-SAF, and SACCF-PF, where 417, 425,
- and 247 Argo profiles in eddy interiors were detected, respectively.

### 2.2 Eddy detection, tracking and CFE categorization

- We combined the Okubo-Weiss (OW) parameter method with the outermost closed contour of SLA to detect
- eddies. As a widely used eddy detection method, the OW parameter method was developed based on flow field
- deformation by high vorticity or high strain (Okubo, 1970; Weiss, 1991). The OW parameter is defined as:

$$W = s_n^2 + s_s^2 - \omega^2, (2)$$

- where  $s_n = \frac{\partial u'}{\partial x} \frac{\partial v'}{\partial y}$  and  $s_s = \frac{\partial v'}{\partial x} + \frac{\partial u'}{\partial y}$  are the normal and shear components of strain, respectively, and
- $\omega = \frac{\partial v'}{\partial x} \frac{\partial u'}{\partial y}$  is the relative vorticity of the flow. The sign of W determines a region to be strain-dominated (W > 0)
- or vorticity-dominated (W 

 $\sum_{i=1}^{N} EKE_i \cdot ds$ , where  $EKE_i$  is the EKE for grid i, N is the grid amount within an eddy, ds is the grid area. The eddy nonlinearity parameter ( $\beta$ ) was computed based on  $\beta = U/C$ , where U is the maximum circum-average geostrophic velocity within the eddy, and C represents the eddy's transporting speed (Chelton et al., 2011). The eddy is nonlinear when  $\beta > 1$ , indicating the presence of trapped fluid parcels advected with the eddy movement.

While climatological fronts define the ACC's mean structure (Park et al., 2019), their positions exhibit meridional variability influenced by both bathymetry and eddy activity (Kim and Orsi, 2014; Thompson et al., 2010): Fronts stabilize over major bathymetric features (e.g., the Pacific-Antarctic Ridge) but show maximum variability in flat basins, with widened frontal zones developing downstream of obstacles like the Campbell Plateau. To account for topographically induced frontal displacements, we defined frontal zones as a strap  $\pm 15$  km expanded in the normal directions from each climatological front, consistent with observed SO frontal oscillation area (Kim and Orsi, 2014). Thereby, we classified CFE dynamics into three phases: pre-cross-frontal (approaching the frontal zone), cross-frontal (within the frontal zone), and post-cross-frontal (exiting the frontal zone). For statistical analysis, CFEs were categorized into four types: (1) Front-generated eddies, (2) Front-dissipated eddies, (3) Transient frontal eddies (both generated and dissipated within the same frontal zones), and (4) Complete CFEs (undergoing all three phases). Accordingly, front-generated eddies propagating away (type 1–3) and eddies

propagating into the frontal zones and dissipating there (type 2-3) were identified as partial CFEs.

The Rich States in State | Sta

Figure 1. Study region. (a) Sea Level Anomaly (*SLA*) distribution on January 20<sup>th</sup>, 2022. (b) Spatial distribution of mean eddy kinetic energy (*EKE*) during 2000–2022. Thick colored lines from north to south represent the northern boundary of ACC (NB), the Subantarctic Front (SAF), the Polar Front (PF), the Southern Antarctic Circumpolar Current Front (SACCF) and the southern boundary of ACC (SB; Park et al., 2019). Significant seafloor topographies have been labeled, with UFZ denoting the Udintsev Fracture Zone.

10

#### 3 Results

#### 3.1 Analysis of CFE characteristics

CFE transport is active at all ACC fronts in the Pacific sector (Figure 2). Equatorward-moving CFEs 184 185 consistently outnumber poleward-moving eddies at each front (Figure 2b). CEs dominate equatorward CFEs, with 186 CEs outnumbering AEs by a factor of ≥1.5, while AEs prevail in poleward CFE motions. The resulting hierarchy of CFE prevalence is as follows: equatorward CEs are the most frequent (36% of total CFEs), followed by poleward 187 188 AEs (27%) and equatorward AEs (21%), with poleward CEs (16%) being the least frequent. Among different 189 fronts, the SAF hosts the most eddy occurrences (26% of total CFEs), followed by the PF (25%), reflecting intense 190 eddy-mean flow interactions around these two fronts. The NB (21% of total CFEs) and the SACCF (20%) exhibit 191 comparable and moderate CFE levels, while the southernmost SB (11%) displays the lowest CFE exchanges. 192 The frontal system exhibits strong meandering patterns due to topographic steering, accompanied by spatially 193 heterogeneous CFE distributions. CFE occurrence peaks downstream of prominent topographic features, 194 particularly near the Campbell Plateau (150°E-180°E; 39% of total CFEs) and downstream of the Udintsev 195 Fracture Zone (125°W-160°W; 38%), where multiple fronts converge (Figure 2a). Eddies may cross multiple 196 fronts sequentially at these frontal convergent regions. The majority of eddies cross a single front (Figure 2c). 197 Double-frontal crossings (total 434) occur preferentially at southern fronts (SACCF/SB; > 50% of cases; Figure 2d). 198 Triple-frontal crossings are rare and primarily limited to the PF/SACCF/SB system (Figure 2e), and no instances 199 of quadruple-frontal crossings were observed. 200 Consistent with ACC dynamics, both cross-frontal CEs and AEs predominantly propagate eastward (> 70% of 201 cases; Figure 3), contrasting with typical Rossby wave-driven westward-propagating mesoscale eddies in other 202 ocean basins (Frenger et al., 2015). Long-distance propagating eddies are concentrated near the SAF and PF, with 203 CEs being the main contributors. The majority of CEs propagate northward (over 64% of all CEs), with maximum 204 displacements reaching approximately 8° latitude. AEs are more inclined to move southward (over 54% of all AEs), 205 with southward displacements confined within 6° latitude. Percentages in Figure 3 show that short-distance 206 movements (within 2°) are more frequent for AEs at each front. The CEs' dominance in long-distance transport 207 indicates their greater energetics compared to AEs. These patterns highlight how ACC-steered eddy motions 208 facilitate distinct meridional exchange pathways, with CEs playing a disproportionate role in long-distance 209 transport, particularly at the major frontal jets. 210 Most CFEs exhibit nonlinear characteristics, with 98% classified as nonlinear (Figure 3k), confirming their

capability to trap water mass. In the nonlinearity regime ( $\beta > 1$ ), equatorward-moving CEs constitute 70% of the

11

total cross-frontal CEs, and 59% of the total cross-frontal AEs are poleward-moving ones (not shown). In the high nonlinearity regime ( $\beta > 5$ ), the proportion of CEs is notably higher than AEs, consistent with the greater dynamic vigor of CEs observed in the above analyses. Therefore, the cross-frontal transport achieved by eddies, primarily equatorward-moving CEs and poleward-moving AEs, can facilitate the redistribution of distinct source water masses and mitigate thermohaline gradients across frontal zones. Quantitative analysis of CFE types reveals distinct latitudinal patterns in eddy behavior (Table 1). Less than half (25.65-45.68%) complete full frontal crossing, with success rates decreasing poleward (44.58% at NB, 45.68% at SAF vs. 25.65% at SB). CEs consistently outperform AEs (maximum 5.57% difference at PF). Over half of the eddies undergo generation or dissipation within frontal zones, and this proportion increases towards higher latitudes. Among the front-generated eddies, AEs slightly outnumber CEs at SAF and PF, while CEs dominate at the other fronts. Dissipation follows similar patterns, where AEs outnumber CEs only at SAF. Transient frontal eddies also occur more frequently at the higher-latitude fronts, with CEs generally exceeding AEs (except at SAF and PF). The latitudinal differences in eddy behavior, particularly the declined complete CFEs but the enhanced partial and transient frontal crossing eddies in weaker fronts (e.g., SACCF, SB), highlight the critical role of frontal jet instability in modulating eddy generation, transport efficiency, and cross-frontal exchange. Cross-frontal CEs and AEs show similar distributions in lifespan, propagation distance, and size (Figure 4). Both types show a steep decline in abundance with increasing lifespan. Eddies with lifespans ≤ 50 days dominate, constituting 68% of the total eddy population, while only 2% exceed 200 days (Figure 4a). Propagation distances are confined predominantly to ≤ 300 km (60% of total CFEs). CEs slightly outnumber AEs at longer distances (300-1000 km; Figure 4b). Size distributions reveal that ~70% of the total sample have mean radii of 30-50 km (Figure 4c). Notably, CEs dominate at smaller radii (< 50 km), while AEs prevail among larger eddies. This distribution pattern is consistent with maximum radius statistics (Figure 4d). These CFE characteristics align with previously reported eddies in the Pacific sector (Duan et al., 2016). Front-generated or dissipated eddies typically exhibit relatively short lifespans (lifespan ≤ 30 days: 62% of generated eddies, 62% of dissipated eddies), limited propagation distances (predominantly confined to ≤ 200 km: 61% generated, 59% dissipated), and small radii (mean radii ≤ 35 km: 51% generated, 52% dissipated; maximum radius ≤ 50 km: 67% generated, 65% dissipated), with very few exceptions in high-value parameter ranges (Figure 4a-d). Conversely, completely transported CFEs display marked longer lifespans (lifespan > 30 days: 72% of the complete CFEs), greater propagation distances (> 200 km: 82%), and larger radii (mean radii > 35 km: 77%;

maximum radii > 50 km: 68%). It's notable that CEs significantly outnumber AEs among small-scale completely

254255

257258

264265

267268

12

transported CFEs (mean radii: 20–40 km; maximum radii < 60 km), with 34% versus 24%. The higher abundance of small-scale CEs suggests their efficiency in cross-frontal transport processes. Over the 22 years, both poleward- and equatorward-moving eddies show pronounced interannual variability

Over the 22 years, both poleward- and equatorward-moving eddies show pronounced interannual variability (Figure 4e). The annual abundance hierarchy, equatorward CEs > poleward AEs > equatorward AEs > poleward CEs, mirrors the total distribution in Figure 2b and is primarily constituted by completely transported CFEs (Figure 5d), underscoring the enhanced capacity of equatorward CEs and poleward AEs for sustained cross-frontal propagation. In contrast, partial or transient CFEs exhibit no clear polarity preferences, although CEs are slightly dominant equatorward types and AEs prevail among poleward types in most years (Figure 5a–c).

The CEs consistently exhibit approximately 1.5-fold greater  $EKE_T$  than AEs (Figure 4f). While CEs'  $EKE_T$ shows a non-significant increasing trend  $(1.52 \pm 1.66) \times 10^6 \text{ m}^4\text{s}^{-2}\text{yr}^{-1}$ , p = 0.08), this result is influenced by an anomalously low 2017 value that coincides with an EKE minimum reported by Fu et al. (2023) in the central Pacific sector. When excluding this outlier, CEs'  $EKE_T$  trend becomes statistically significant with  $(1.98 \pm 1.53) \times$  $10^6 \text{ m}^4 \text{s}^{-2} \text{yr}^{-1}$  (p = 0.02). AEs' *EKE<sub>T</sub>* displays a significant increasing trend of (1.58 ± 0.74) ×  $10^6 \text{ m}^4 \text{s}^{-2} \text{yr}^{-1}$  (p < 0.001). These results indicate that  $EKE_T$  for both polarity eddies increases over the 22 years, with CEs exhibiting greater interannual variability. As established in Figure 2, equatorward-propagating CEs and poleward-propagating AEs dominate cross-frontal eddy abundance. Examining their EKE<sub>T</sub> trends reveals that they exhibit substantially stronger signals compared to those for the overall CFE population, with increasing trends for equatorward CEs of  $(2.27 \pm 2.11) \times 10^6 \text{ m}^4 \text{ s}^{-2} \text{ yr}^{-1} \text{ (p} = 0.03)$  and poleward AEs of  $(1.51 \pm 1.10) \times 10^6 \text{ m}^4 \text{s}^{-2} \text{yr}^{-1} \text{ (p} = 0.01)$ . More specifically, the observed increase in annual mean  $EKE_T$  is primarily driven by complete CFEs, with CEs and AEs showing significant trends of  $(2.86 \pm 2.42) \times 10^6 \text{ m}^4\text{s}^{-2}\text{yr}^{-1}$  (p = 0.02) and  $(1.93 \pm 1.25) \times 10^6 \text{ m}^4\text{s}^{-2}\text{yr}^{-1}$  (p < 0.01), respectively (Figure 5h). In contrast, partial or transient CFEs show lower  $EKE_T$  levels and no significant trends (Figure 5a-c), with the exception of transient frontal CEs, which also exhibit an increasing trend of (2.34 ± 2.07) ×  $10^6 \,\mathrm{m}^4 \,\mathrm{s}^{-2} \mathrm{yr}^{-1}$  (p = 0.02). The energy contrast between complete CEs and complete AEs is more pronounced than in other eddy types. The intensification pattern is consistent in mean EKE data (Figure S1 in Supplementary Materials), which shows a greater enhancement in CEs. These results indicate the rise in CFE activity from 2000-2022 is largely attributable to equatorward-propagating complete CEs and poleward-propagating complete AEs, the most energetic two types. Notably, the time series demonstrates a decoupling between EKE intensity and eddy abundance, exemplified by 2017 when CE abundance exceeded AE, yet their EKE<sub>T</sub> dropped below AE levels (Figure 4e, f).

Figure 2. CFE trajectories and statistics of eddy counts. (a) CFE trajectories. The color along each front represents the relative occurrence of CFEs (%) per 5° latitudinal bin. The framed regions denote the area where active CFE activity occurs. (b) Total number of CFEs, divided into types of equatorward and poleward directions; (c)—(e) Numbers of single-, double-, and triple-frontal crossing CFEs, respectively. Red represents anticyclonic eddies (AEs) and dark blue denotes cyclonic eddies (CEs).

Figure 3. Relative movement trajectories of CFEs (a–j) and the percentage distribution of eddy nonlinearity  $\beta$  (k). In (a–j), black percentages represent the proportion of eddies moving in different quadrant directions calculated based on the end point of the trajectory and purple percentages indicate the proportions of eddies with movement distances within 2° range, with the coordinate origin (0°, 0°) denoting the eddies' generation locations. Note that eddies crossing multiple fronts may appear repeatedly at different frontal positions in this analysis.

## 281 Table 1. Proportions of numbers of different eddy types relative to the total number of CFEs at each frontal zone.

| Type                  | Eddy polarity | NB     | SAF    | PF     | SACCF  | SB     |
|-----------------------|---------------|--------|--------|--------|--------|--------|
| (1) Front-generated   | AE            | 16.04% | 18.28% | 19.61% | 25.65% | 26.11% |
| eddies                | CE            | 17.02% | 16.03% | 19.03% | 26.46% | 27.70% |
| (2) Front-dissipated  | AE            | 17.31% | 17.13% | 18.88% | 21.54% | 24.71% |
| eddies                | CE            | 18.88% | 16.45% | 20.27% | 25.94% | 26.68% |
| (3) Transient frontal | AE            | 6.82%  | 7.07%  | 9.03%  | 12.02% | 14.17% |
| eddies                | CE            | 7.01%  | 6.50%  | 7.87%  | 14.30% | 16.68% |
| (4) Complete CFEs     | AE            | 21.27% | 21.66% | 16.77% | 12.06% | 12.64% |
|                       | CE            | 23.31% | 24.02% | 22.34% | 14.67% | 13.01% |

Figure 4. Statistical characteristics of different types of CFEs (a–d) and time series of annual CFE counts (e) and annual mean  $EKE_T$  (f). Eddy counts according to (a) eddy lifespan, (b) propagation distance, (c) mean radius over lifespan, (d) maximum radius in the lifespan. Note the x-axis in a–b are not equidistant at higher values. In (a–d), "All" represents all CFEs, "Generated" denotes front-generated eddies, "Dissipated" indicates front-dissipated eddies, and "Transported" shows complete CFEs. In (f), the annual mean  $EKE_T$  for all AEs and CEs are depicted by light red and light blue solid lines, respectively, with their linear trends indicated by solid lines in the same colors. Superimposed are the extracted subsets of poleward-moving AEs and equatorward-moving CEs, depicted by light red and light blue dashed lines, with their linear trends shown by dashed lines in the same respective colors. Error shadings represent one standard deviation, and slope values are given with  $\pm 95\%$  confidence intervals.

Figure 5. Statistical characteristics of four types of CFEs. (a–d) Time series of annual counts for (a) Front-dissipated eddies, (b) Front-generated eddies, (c) Transient frontal eddies, and (d) Complete CFEs. (e–h)annual mean  $EKE_T$  for (e) Front-dissipated eddies, (f) Front-generated eddies, (g) Transient frontal eddies, and (h) Complete CFEs. The dashed lines in (e–h) show the linear trends, with colors matching their respective time series. The slope values of the trends are provided with  $\pm 95\%$  confidence intervals.

#### 3.2 Variations in $EKE_T$ of complete CFEs during frontal crossing

While only a small fraction of eddies complete full cross-frontal transport (Table 1, Figure 5), these energetic eddies, originating in non-frontal zones and crossing entire frontal boundaries, likely dominate long-distance heat and material exchange. To quantify the influence of eddy-jet interactions on EKE in this type during frontal crossing, we analyzed the evolution of  $EKE_T$  across the three phases: pre-crossing, crossing, and post-crossing. The Southern Hemisphere's intrinsic vorticity asymmetry (clockwise CEs vs. counterclockwise AEs) creates fundamental polarity differences in energy exchange when interacting with eastward frontal jets. Hence, eddies of opposing polarities and directions are expected to exhibit distinct patterns of  $EKE_T$  variation during the cross-frontal transport.

As shown in Figures 6 and 7, complete CFEs at the northern fronts (NB, SAF, PF) exhibit significantly higher  $EKE_T$  values than at the southern fronts (SACCF, SB), which is consistent with the previous result of greater  $EKE_T$  of complete CFEs (Figure 5) and their often occurrence at the northern fronts (Table 1). For instance, mean

higher  $EKE_T$  values than at the southern fronts (SACCF, SB), which is consistent with the previous result of greater  $EKE_T$  of complete CFEs (Figure 5) and their often occurrence at the northern fronts (Table 1). For instance, mean  $EKE_T$  at SAF (2.78 × 10<sup>8</sup> m<sup>4</sup>/s<sup>2</sup>) is 2.22 × 10<sup>8</sup> m<sup>4</sup>/s<sup>2</sup> higher than that at SB (5.62 × 10<sup>7</sup> m<sup>4</sup>/s<sup>2</sup>). Poleward-moving AEs consistently gain kinetic energy during frontal crossing (5.87–41.83% increase), with further post-crossing amplification (14.97–88.77%), indicating sustained energy extraction from mean flows (Figure 6, Table 2). In contrast, poleward CEs generally lose energy at the northern fronts (39.51–59.62% reduction in the post-crossing phase), though they show atypical 19.57–29.55%  $EKE_T$  increases at the southern two fronts, but subsequently decline post-crossing (14.72% and 19.60% decreases, respectively).

Equatorward-moving CFEs exhibit opposing  $EKE_T$  behaviors (Figure 7, Table 2): AEs consistently lose energy, showing 4.96–18.75% reduction during frontal crossing and further decline post-crossing (29.88–65.71% reduction), while CEs generally gain energy (29.82–48.86% increase post-crossing at northern fronts). This pattern reverses at southern fronts (SACCF, SB), where CEs show initial  $EKE_T$  gains during crossing but subsequent losses (e.g., 14.36% decrease post-SACCF crossing), mirroring poleward CEs' behavior.

The observed polarity asymmetry yields systematically higher post-crossing  $EKE_T$  for poleward AEs and equatorward CEs, demonstrating a clear polarity- and direction-dependent energy transfer during eddy-jet interactions. While poleward AEs efficiently extract energy from frontal jets, equatorward AEs consistently dissipate energy during and after frontal crossing. Similarly, equatorward CEs gain substantial energy, whereas poleward CEs lose energy at the northern three fronts. The anomalous transient energization of CEs at the SACCF and SB, followed by eventual decay, likely reflects the weaker dynamical and stronger hydrographic dynamic characteristics of these southernmost fronts (Park et al., 2019; Thorpe et al., 2002; Vereshchaka et al., 2021).

19

Table 2. Changes in mean  $EKE_T$  during different phases for complete cross-frontal eddies (CFEs) relative to pre-crossing values (+: increase; -: decrease). Crossing phases represent when eddies are in the frontal zones, while post-crossing phases indicate when eddies are moving away from the frontal zones.

| Direction          | Eddy<br>polarity | Phase    | NB      | SAF     | PF      | SACCF   | SB      |
|--------------------|------------------|----------|---------|---------|---------|---------|---------|
|                    | AE -             | crossing | +41.03% | +5.87%  | +34.80% | +41.83% | +34.98% |
| poleward-moving —  | AL -             | post     | +63.83% | +14.97% | +59.15% | +52.94% | +88.77% |
| poiewaru-moving    | CE -             | crossing | -4.67%  | -10.12% | -4.96%  | +19.57% | +29.55% |
|                    | CE               | post     | -39.51% | -59.62% | -46.85% | -14.72% | -19.60% |
|                    | AE -             | crossing | -15.29% | -4.96%  | -9.98%  | -18.75% | -9.83%  |
|                    | AL -             | post     | -47.39% | -29.88% | -42.67% | -65.71% | -35.74% |
| equatorward-moving | CE -             | crossing | +27.98% | +35.97% | +22.86% | +3.64%  | +38.01% |
|                    | CE -             | post     | +36.93% | +48.86% | +29.82% | -14.36% | +5.04%  |

Figure 6. Probability density function (PDF) of  $EKE_T$  for poleward-moving CFEs in pre-crossing, crossing, and post-crossing phases. Dashed lines indicate median  $EKE_T$  values. Blue and red colors represent CEs and AEs, respectively.

Figure 7. Probability density function (PDF) of  $EKE_T$  for equatorward-moving CFEs in pre-crossing, crossing, and post-crossing phases. Dashed lines indicate median  $EKE_T$  values. Blue and red colors represent CEs and AEs, respectively.

341342

348349

358359

21

#### 3.3 Thermohaline transport effects of CFEs

Argo  $\theta$ -S profiles (Figure 8) demonstrate that marked meridional thermohaline gradients exist between eddy intervals in different interfrontal zones, with colder and fresher water properties poleward. In the same zones, CEs consistently exhibit colder, fresher properties with shallower isopycnals (upper 1000 dbar), while AEs contain warmer, saltier waters with deeper isopycnals, reflecting their respective meridional origins. These polarity-dependent hydrographies underscore how nonlinear eddies mediate cross-frontal exchange. Nevertheless, some SACCF-PF AEs trapped anomalously cold, fresh polar waters ( $\theta_{min} = -1.76$ °C and S < 34.0 psu), similar to the water properties observed in some AEs in the SB-SACCF zone, indicating equatorward transport of polar waters by AEs. It's noteworthy that eddy-induced upwelling (CEs) and downwelling (AEs) achieve vertical displacements but do not alter isopycnal properties in source water columns (Falkowski et al., 1991; Li et al., 2022). Thus, this mechanism can only explain some overlapping  $\theta$ -S signatures in vertically displaced water columns between CEs and AEs within the same interfrontal zones. An analysis of radius-normalized  $\theta$  and S distributions reveals distinct water mass signatures in CEs and AEs across northern interfrontal zones (SAF-NB, PF-SAF, SACCF-PF). In the SAF-NB region, well-defined Subantarctic Mode Water (SAMW), Antarctic Intermediate Water (AAIW), and Upper Circumpolar Deep Water (UCDW) layers (Table 3) are observed from upper to lower layer in the AE (Figure 9d, j), confirming their local origin within the Antarctic Convergence Zone. Conversely, the CE in the same region shows markedly different thermohaline structures (Figure 9a, g), with upper layers (<1000 dbar) lacking SAMW/AAIW signatures and instead containing colder, fresher waters of southern origin. Neutral density  $(\gamma^n)$  surfaces in the CE appear 200–300 dbar shallower than in the AE, demonstrating that CEs effectively transport high-potential-energy southern waters into the SAF-NB zone. This creates strong mesoscale potential energy contrasts between the low-potential-energy waters in AEs and the high-potential-energy waters in CEs. These contrasts provide an energetic basis for baroclinic instability through the release of available potential energy (Fu et al., 2023). In the PF-SAF region, both the CE and AE maintain thermohaline anomalies similar to those in the SAF-NB zone but with reduced magnitude, preserving the characteristic warmer/saltier AE and colder/fresher CE signatures (the middle panels of Figure 9). Notably, only the CE's upper layer exhibits distinct Winter Water (WW) characteristics, confirming their southern origins. Below this, the normalized CE sequentially displays UCDW and LCDW, while the AE shows only UCDW beneath the relatively warm and salty Antarctic Surface Water within the upper 2000 dbar. The vertical isopycnal structure reveals depth-dependent displacements: in the near-surface layer, the CE's isopycnal  $y^n$ =27.1 kg/m<sup>3</sup> is ~100 dbar shallower than the AE, while at intermediate depths, the CE's

isopycnal  $\gamma^n$ =27.6 kg/m<sup>3</sup> (400–600 dbar) is ~500 dbar shallower than the AE (~1000 dbar).

The thermohaline anomalies between CE and AE still exist in the SACCF-PF zone (the right panels of Figure 9). In the CE, a subsurface WW layer overlies a warm UCDW core, with LCDW dominating below 1000 dbar, showing a characteristic of waters south of the SACCF (Aoki et al., 2013; Auger et al., 2021). While the AE also contains these water masses, they show weaker WW expression, a more pronounced  $\theta_{max}$  core, and vertically extended UCDW, reflecting their relatively northern origins. Isopycnals in the CE remain consistently 300–400 dbar shallower than in the AE throughout the water column.

Therefore, the normalized AE and CE possess distinct water mass distributions within the same inter-frontal zones, marked by profound isopycnal thermohaline differences. AEs and CEs transport their respective source water masses into these zones, amplifying mesoscale hydrographic variability. The above comparative analysis demonstrates that cross-frontal CEs play a dominant role in meridional water mass transport, particularly in the SAF-NB and SACCF-PF zones, consistent with their greater dynamical vigor. This cross-frontal exchange reduces baroclinicity between interfrontal zones while enhancing mesoscale available potential energy within individual zones.

Figure 8. Potential temperature-salinity (θ-S) diagrams in the eddy interiors observed in different inter-frontal zones. (a)
 SAF-NB; (b) PF-SAF; (c) SACCF-PF; (d) SB-SACCF.

24

# Table 3. Criteria for the division of water masses according to potential temperature ( $\theta$ , °C), salinity (S, psu) and neutral density ( $\gamma^n$ , kg/m³).

| Water mass | θ (°C)              | S (psu)      | $\gamma^n (kg/m^3)$ | Reference                         |
|------------|---------------------|--------------|---------------------|-----------------------------------|
| SAMW       |                     | 34.35–34.60  | 26.50–27.10         | Carter et al., 2022               |
| AAIW       |                     | $S_{\min}$   | 27.10-27.60         | Xia et al., 2022;                 |
|            |                     | 34.15-34.30  |                     | Valla et al., 2018                |
| UCDW       | $	heta_{	ext{max}}$ |              | 27.55–28.00         | Naveira Garabato et al., 2002     |
| LCDW       |                     | $S_{ m max}$ | 28.00-28.26         |                                   |
| WW         | $	heta_{	ext{min}}$ |              | 27.20–27.40         | Azarian et al., 2024; Fischer and |
|            | -0.55-3.00          |              |                     | Visbeck, 1993                     |

\*SAMW, Subantarctic Mode Water; AAIW, Antarctic Intermediate Water; UCDW, Upper Circumpolar Deep Water; LCDW, Lower Circumpolar Deep Water; WW, Winter Water.

Figure 9. Sectional distributions of  $\theta$  (a=f) and S (g=l) along normalized eddy radius (R) direction in the inter-frontal zones of SAF-NB (the left panels), PF-SAF (the middle panels), and SACCF-PF (the right panels). Black thick contours indicate neutral density ( $\gamma^n$ , kg/m³), thin contours represent  $\theta$  or S isolines, respectively.

26

#### 4 Discussion

Pacific sector, characterized by three key aspects: (1) a distinct abundance hierarchy among CFE types, (2) 389 contrasting EKE intensities and trends, and (3) polarity and direction-selective energy transfers during eddy-jet 390 interactions. The equatorward-moving CEs dominate CFE activity (36% of total CFEs), followed by poleward AEs 391 (27%), and then equatorward AEs and poleward CEs. This pattern, in which CEs predominantly migrate 392 equatorward and AEs poleward, is consistent with established eddy dynamics in the SO (He et al., 2023; Li et al., 393 2022; Patel et al., 2019). The dominant types exhibit superior energetic characteristics with higher EKE (Figures 4f, 394 S1), longer propagation distances (Figure 3), and stronger nonlinearity, compared to their counterparts. Critically, 395 while CFE abundance shows no significant trend over 2000-2022, both polarity groups experienced substantial  $EKE_T$  intensification, with CEs gaining energy at  $(1.98 \pm 1.53) \times 10^6$  m<sup>4</sup>s<sup>-2</sup>yr<sup>-1</sup> (excluding the anomalously low 396 2017 value) and AEs at  $(1.58 \pm 0.74) \times 10^6 \text{ m}^4\text{s}^{-2}\text{yr}^{-1}$ . This enhancement is primarily driven by 397 398 equatorward-propagating complete CEs and poleward-propagating complete AEs, contrasting with partial and 399 transient CFEs which show no significant trends (Figures 4, 5). Non-CFEs (both CEs and AEs) also exhibited no 400 comparable  $EKE_T$  or EKE trends (Figures 10, S2). This evidence reveals the preference for wind stress driving elevated EKE (Hogg et al., 2015; Menna et al., 2020), demonstrating that enhanced wind stress preferentially 401 402 energizes cross-frontal activity, especially the CFEs achieving complete frontal crossing. The predominance of 403 equatorward CEs aligns with intensified Ekman transport patterns reported by Shi et al. (2025), also suggesting 404 wind-driven facilitation of meridional eddy migration. 405 Building on Fu et al.'s (2023) framework of wind-driven energy pathways (baroclinic: mean kinetic energy→ 406 mean available potential energy → mesoscale available potential energy → EKE; barotropic: mean kinetic energy → 407 EKE), we demonstrated that cross-frontal eddy-jet interactions exhibit polarity- and direction-dependent energy 408 transfers (as illustrated in Figure 11). Equatorward complete CEs and poleward complete AEs gain substantial 409 kinetic energy from jets (e.g., +48.86% post-SAF crossing and +63.83% post-NB crossing, respectively), 410 potentially through two synergistic mechanisms: (1) barotropic instability from enhanced horizontal shear when 411 eddy rotation aligns with the eastward jet (Qiu et al., 2024), and (2) baroclinic instability triggered by potential 412 energy release for enhanced hydrographic gradients with ambient waters (Fu et al., 2023). These kinetic energy 413 gains during cross-frontal activity dynamically fuel these types of eddies, and subsequently sustain long-distance 414 propagation and meridional heat and material transport (He et al., 2023; Patel et al., 2019; Sun et al., 2019). Conversely, poleward CEs and equatorward AEs show significant energy losses (e.g., -59.62% and -29.88% 415

This study reveals a fundamental polarity- and direction-dependent asymmetry in CFEs dynamics within the

431432

27

post-SAF crossing), possibly due to counter-rotational turbulent dissipation (Dong et al., 2017; Jan et al., 2017) and upwelling (downwelling)-induced baroclinicity reduction with the ambient waters. The intensifying and poleward-shifting westerlies have emerged as the dominant dynamic forcing mechanism in the SO (Behrens and Bostock, 2023; Hogg et al., 2015). Meanwhile, buoyancy forcing due to meridionally inhomogeneous ocean warming has been proved responsible to ACC acceleration at 48°S-58°S (Shi et al., 2021). Our results demonstrated that CFEs play a vital role in mediating the oceanic response through compensating heat transport. By facilitating poleward warm-water transport via AEs and equatorward cold-water transport via CEs, CFEs mitigate cross-frontal water mass property gradients, effectively buffering wind- or warming-induced baroclinicity increases. This eddy-mediated regulation maintains the SO's thermal equilibrium and modulates the ACC's response to external forcing, highlighting CFEs' dual role as both energy transporters and dynamical stabilizers in a changing climate. In addition, the multiple-front-crossing eddies warrant attention. These eddies (12% of total; Figure 2c-e) demonstrate exceptional transport capacity, with apparently farther meridional propagation than single-front-crossing eddies (mean 2.24° vs. 0.78°). Their extended lifespans and capacity of long-distance propagation (Figure 12) make them highly efficient biogeochemical transporters across fronts. A southward shift in westerlies may push their generation zones poleward (Shi et al. 2025), potentially enhancing their thermal impacts on Antarctic ice shelves through amplified heat delivery. It should be noted that this study defines each frontal zone as a 30 km-wide strip-shaped area but does not account for potential interannual or seasonal variations that may extend beyond this range. Similarly, all qualified

Argo profiles from 2000 to 2022 were used without considering interannual or seasonal variability in hydrographic

properties. These limitations inevitably introduce certain uncertainties.

Figure 10. Time series of annual mean  $EKE_T$  for eddies in the interfrontal zones.  $EKE_T$  is shown by blue solid line for CEs and red solid line for AEs, with linear regression indicated by dashed lines, error shadings representing one standard deviation, and slope values given with  $\pm$  95% confidence intervals.

Figure 11. Illustrations of EKE variations during frontal crossing for poleward AEs and equatorward CEs (modified from

Figure 1 in Chapman et al., 2020). The thickness of rotational velocity vectors represents relative flow intensity.

Figure 12. Proportional distributions lifespan (a), propagation distance (b), mean radius (c) and max radius (d) for

multi-front-crossing eddies (solid lines) versus single-front-crossing eddies (dashed lines).

31

#### **5 Conclusions**

This study reveals a fundamental polarity- and direction-dependent asymmetry in the behavior of cross-frontal eddies (CFEs) in the SO's Pacific sector, with primary implications for heat transport and climate 446 447 regulation. A clear hierarchy exists that CEs are the most prevalent and energetic (36% of total CFEs), followed by 448 AEs (27%). These two types dominate long-distance cross-frontal transport and have significantly intensified 449 over the past two decades (2000-2022), gaining kinetic energy from the frontal jets during crossing. This trend 450 is absent in eddies that partially or transiently cross fronts, or do not cross fronts, showing that wind 451 stress intensification preferentially energizes complete cross-frontal activity. A dual mechanism could explain the energy gains in the equatorward CEs and poleward AEs that (1) barotropic instability from enhanced horizontal 452 453 shear due to aligned rotation with eastward jets and (2) baroclinic instability from the potential 454 energy release created by sharp hydrographic contrasts with surrounding waters. Conversely, counter-rotational 455 eddies experience eddy dissipation. CFEs act as efficient transporters of distinct water masses: CEs trap and carry cold, fresh southern waters equatorward, while AEs transport warm, salty northern waters poleward. This process 456 457 directly mitigates the sharp meridional gradients. By compensating for wind-driven increases and 458 inhomogeneous ocean warming in baroclinicity, CFEs help maintain the SO's thermal equilibrium and modulate 459 the ACC's response to climate change. 460 These results advance our understanding of cross-frontal mesoscale processes in the ACC and potentially 461 provide a mechanism-based explanation of the distribution of marine species in the SO, which originate from the 462 regions beyond the Antarctic region. The identified polarity asymmetries and preference for energy transfers 463 have vital implications for parameterizing eddy effects in climate models, particularly under projected wind regime 464 shifts.

Data availability

485

486 487

32

## 466 The satellite altimeter data are available online at the Copernicus Marine Service (Global Ocean Gridded L4 Sea Surface Heights And Derived Variables at https://marine.copernicus.eu/). The frontal data used in this study 467 468 were sourced from Park et al. (2019) at https://doi.org/10.17882/59800. Argo profiles are available at the website of 469 http://www.argo.net. 470 **Author contributions** 471 Huimin Wang: Methodology, Software, Formal analysis, Investigation, Data curation, Writing-original draft, 472 Visualization. Lingqiao Cheng: Conceptualization, Methodology, Resources, Writing-original draft, 473 Writing-review & editing, Supervision, Project administration, Funding acquisition. Erik Behrens: Validation, 474 Writing-review & editing. Zhuang Chen: Validation, Writing-review & editing. Jennifer Devine: Writing-review & 475 editing. Guoping Zhu: Resources, Writing-review & editing, Funding acquisition. 476 **Competing Interests** 477 The authors declare that they have no conflict of interest. 478 Acknowledgments 479 We thank the Key Laboratory of Sustainable Exploitation of Oceanic Fisheries Resources, Shanghai Ocean 480 University and the Joint Research Center on Antarctic Marine Science supported by Shanghai Ocean University, 481 Earth Sciences New Zealand, and the University of Otago for their support and the utilization of their laboratory 482 facilities. 483 Financial support This work was funded by the National Key Research and Development Program of China (grant no. 484

2023YFE0104500), the National Natural Science Foundation of China (grant no. 42130402), Shanghai Top-tier

Talent Program of Eastern Talent Plan (grant no. BJKJ2024059), and supported by the Catalyst Fund from

Government funding, administered by the Royal Society Te Apārangi.

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

- structuring of the Subantarctic myctophid Electrona carlsbergi in the Antarctic Circumpolar Current and
- Antarctic Slope Current off the South Shetland Islands. Palaeogeography, Palaeoclimatology, Palaeoecology,
- 675: 113062.