# Peer review of "Polarity and direction dependence of energetic cross-frontal eddy transport in the Southern"

_EGUsphere, 2025_

## Referee Comment (RC1)

Review of the manuscript egusphere-2025-5038 **"Polarity and direction dependence of energetic cross-frontal eddy transport in the Southern Ocean's Pacific sector"** by Huimin WANG, Lingqiao CHENG, Erik BEHRENS, Zhuang CHEN, Jennifer DEVINE, and Guoping ZHU submitted to *EGUsphere*

**Reviewer: Igor M. Belkin**
**Date: December 14, 2025**

**Summary:** This study is methodologically flawed to the point where I must recommend rejection. I do not see how these flaws (enumerated below) can be fixed.

**Major issues:**

**Frontal pattern of the SO:**
Line 49: "In the SO, the transition from the warm subtropical waters to the cold Antarctic waters is not smooth but concentrated along a series of fronts (Deacon, 1937)."
**Comments:** Deacon (1933, 1937) missed the Subantarctic Front (SAF) largely due to the poor spatial resolution of oceanographic observations during RV *Discovery* cruises, with most stations placed three degrees of latitude apart. The SAF was discovered much later. The first definitive report of SAF was published by R.W. Burling in 1961. Thus, regarding the large-scale frontal pattern of the SO, Deacon's works of 1933 and 1937 are outdated. The first modern circumpolar surveys of the SO fronts were published by Orsi et al. (1995) and Belkin and Gordon (1996).

**ACC's northern boundary (NB):**
The authors use the northern boundary (NB) of ACC, which was introduced by Park et al. (2019, pp. 4515-4516): "The streamlines corresponding to the above five circumpolar fronts are determined as follows. First, the ACC northern and southern boundaries are unambiguously defined as those circumpolar streamlines passing through the northernmost and southernmost latitudes of Drake Passage, coinciding with the MDT contour of 0.30 m for the NB and −1.11 m for the SB. There is no great alternative for defining these boundaries because their streamlines are constrained by continental slopes at Drake Passage... ...the definition of the NB from altimetry seems new. Note however that our NB runs close to the northern branch of the SAF (SAF-N) in Sokolov and Rintoul (2009) and Barré et al. (2011). Indeed, it is shown in the following subsection that the newly defined NB from altimetry matches well with the SAF-N identified from hydrography in the study area. Therefore, the altimetry derived NB and the hydrography-derived SAF-N are used interchangeably in this study."
Also, L123: "Park et al. (2019) produced the most updated mapping of the ACC fronts. As shown in Figure 1, from north to south, the key fronts include the northern boundary of ACC (NB), the Subantarctic Front (SAF), the Polar Front (PF), the Southern Antarctic Circumpolar Current Front (SACCF), and the southern boundary of the ACC (SB)."
**Comments:** Fronts are commonly defined as narrow high-gradient zones. The ACC's northern boundary (NB) defined by Park et al. (2019) is a streamline, not a front. Therefore,

the authors should not use NB in their analysis. Traditionally, the northern boundary of ACC is the Subantarctic Front (SAF).

**Eddy tracking:**

L150: "*SLA* data with $0.25°$ spatial resolution were first linearly interpolated to $0.125°$ for better performance in eddy detection."
**Comments:** The interpolation does not mitigate the poor spatial resolution (28 km) of SLA data. Such data is inadequate for studies of mesoscale eddies, whose diameters can be as small as 40-50 km.

L152: "For eddy tracking, the algorithm identifies eddies at time t+1 that meet the following criteria relative to time t: (1) minimal centroid distance, (2) identical polarity (i.e., rotation direction), and (3) the minimum radius variation."
**Comments:** Traditionally, ocean eddies are tracked using their thermohaline (TS) signatures. This is a standard approach successfully utilized in hundreds of field campaigns and data analyses. The authors' ignorance of TS signatures of individual eddies is a major flaw of this study.

L156: "Based on eddy identification and tracking results, this study focuses on eddies with lifespans exceeding 7 days and amplitudes greater than 2 cm."
**Comments:** Mesoscale eddies have a typical lifespan ranging from several weeks up to a few years (rings of major currents). Their typical SSHA amplitudes range widely. In the Southern Ocean, the mean SSHA is 12 cm (Frenger et al., 2015, JGR). In the North Pacific, the SSHA ranges from 10 cm to 30 cm and higher (Ebuchi and Hanawa, 2001, Journal of Oceanography, Fig. 2 and Fig. 3). Rings spawned by major currents have SSHA>40 cm in the Gulf Stream area (Belkin et al., 2020, JPO, Fig. 11), 35-50 cm in young Agulhas rings (van Aken et al., 2003, DSR2; Schmid et al., 2003, DSR2; Wang et al., 2016, JGR) and up to 60 cm in the South Atlantic (Souza et al., 2011, Ocean Science, Table 2). Thus, the 7-day, 2-cm SSH anomalies are hardly qualified as mesoscale eddies. Such anomalies should be considered noise, especially given the accuracy of satellite altimetry of about 1-2 cm.

**Frontal zones:**

L165: "While climatological fronts define the ACC's mean structure (Park et al., 2019), their positions exhibit meridional variability influenced by both bathymetry and eddy activity (Kim and Orsi, 2014; Thompson et al., 2010): Fronts stabilize over major bathymetric features (e.g., the Pacific-Antarctic Ridge) but show maximum variability in flat basins, with widened frontal zones developing downstream of obstacles like the Campbell Plateau. To account for topographically induced frontal displacements, we defined frontal zones as a strap $\pm 15$ km expanded in the normal directions from each climatological front, consistent with observed SO frontal oscillation area (Kim and Orsi, 2014)."
**Comments:** The 15 km threshold grossly underestimates the real observed shifts of SO fronts. Frontal meanders lead to frontal shifts of up to ~100 km in either direction. When a frontal meander pinches off and forms a ring, the ring diameter is typically on the order of 100 km. The 30-km-wide frontal zone definition thus makes little sense.

L433: "…this study defines each frontal zone as a 30 km-wide strip-shaped area but does not account for potential interannual or seasonal variations that may extend beyond this range. Similarly, all qualified Argo profiles from 2000 to 2022 were used without considering interannual or seasonal variability in hydrographic properties. These limitations inevitably introduce certain uncertainties."

**Comments:** In addition to seasonal and interannual shifts of individual fronts, these fronts experience so-called synoptic (or intra-seasonal) variability caused by meanders. Such meanders effectively lead to shifts of individual fronts on the order of 100 km in cross-frontal directions.

**Cross-frontal eddies:**

L193: "CFE occurrence peaks downstream of prominent topographic features, particularly near the Campbell Plateau (150° E–180° E; 39% of total CFEs) and downstream of the Udintsev Fracture Zone (125° W–160° W; 38%), where multiple fronts converge (Figure 2a). Eddies may cross multiple fronts sequentially at these frontal convergent regions. The majority of eddies cross a single front (Figure 2c). Double-frontal crossings (total 434) occur preferentially at southern fronts (SACCF/SB; > 50% of cases; Figure 2d). Triple-frontal crossings are rare and primarily limited to the PF/SACCF/ SB system (Figure 2e), and no instances of quadruple-frontal crossings were observed."

**Comments:** The cross-frontal transport by rings spawned by fronts is well known. However, the cross-frontal transport by other types of mesoscale eddies is a totally different phenomenon, which is exceedingly rare and not well documented from in situ observations. Dufour et al. (2015, JPO) wrote: "However, meridional transport in the Southern Ocean requires crossing multiple intense jets that form the ACC fronts and act as natural barriers to tracer transport. Both observational and experimental studies have demonstrated that mixing across the core of jets is strongly inhibited due to the speed of the jet being higher than the propagation speed of eddies, hence reducing the time during which eddies can stir tracers (e.g., Bower et al. 1985; Lozier et al. 1997; Sommeria et al. 1989; Ferrari and Nikurashin 2010). Nonetheless, strong perturbations, such as rings detaching from the front, might provide a way for tracers to cross the jet cores (Samelson 1992; Wiggins 2005)." –

**Comments:** The above excerpt from Dufour et al. (2015, JPO) makes clear that the cross-frontal eddy transport is likely effectuated by rings, while the cross-frontal transport by other types of mesoscale eddies is exceedingly small except for the cross-frontal transport by subsurface (intrathermocline) lenses at depth (e.g., Bower et al., 2013, DSR2). Thus, the probability of the same eddy crossing multiple fronts is infinitesimally small. Nonetheless, the authors claim that their algorithm detected numerous cases of multiple frontal crossings by the same eddies, with 434 cases of double-frontal crossings, and several cases of triple-frontal crossings (lines 197-198 and Figure 2e). It seems that these numbers are artifacts that resulted from a faulty eddy tracking algorithm.

**Minor edits:**

L55: "Practically" – Delete.

L73: "Electrona calserbgi" should be "Electrona carlsbergi"

L84: "mitigating the eastward flow in the ACC" – "Mitigating" is not the best term in the given context.

L108: "the Campbell Plateau, Pacific-Antarctic Ridge, and Udintsev Fracture Zone (Figure 1)." – Add the Eltanin Fracture Zone to the text and to the maps in Figure 1 and elsewhere.

L112: "during 2000 and 2022" should be "during 2000-2022"

L216: "and mitigate thermohaline gradients across frontal zones." – **Comments:** Replace "mitigate" with a more appropriate term."

L239: "display marked longer lifespans" should be "display markedly longer lifespans"

L284: "Note the x-axis in a‑b are not equidistant at higher values." – Rewrite.

L334: "Argo $\theta$-$S$ profiles (Figure 8) demonstrate that marked meridional thermohaline gradients exist between eddy intervals in different interfrontal zones, with colder and fresher water properties poleward. In the same zones, CEs consistently exhibit colder, fresher properties with shallower isopycnals (upper 1000 dbar), while AEs contain warmer, saltier waters with deeper isopycnals, reflecting their respective meridional origins." – **Comments:** This is a trivial observation: From north to south, the SO temperature and salinity generally decrease across all fronts and interfrontal zones. This general ocean-wide trend has been known for a century.

L423: "CFEs mitigate cross-frontal water mass property gradients" – **Comments:** Avoid using "mitigate" regarding gradients.

L456: "This process directly mitigates the sharp meridional gradients." – **Comments:** Avoid using "mitigate" regarding gradients.

===== END of REVIEW =====